# Evaluation of Thai Silkworm (*Bombyx mori* L.) Hydrolysate Powder for Blood Pressure Reduction in Hypertensive Rats

**DOI:** 10.3390/foods13060943

**Published:** 2024-03-20

**Authors:** Artorn Anuduang, Wan Aida Wan Mustapha, Seng Joe Lim, Somchai Jomduang, Suphat Phongthai, Sakaewan Ounjaijean, Kongsak Boonyapranai

**Affiliations:** 1Department of Food Sciences, Faculty of Science and Technology, Universiti Kebangsaan Malaysia, UKM, Bangi 43600, Selangor, Malaysia; a.anuduang@gmail.com (A.A.); wanaidawm@ukm.edu.my (W.A.W.M.); joe@ukm.edu.my (S.J.L.); 2Innovation Centre for Confectionery Technology, Faculty of Science and Technology, Universiti Kebangsaan Malaysia, UKM, Bangi 43600, Selangor, Malaysia; 3Bio Crenovation Company Limited, 353/2 Moo 9, Tambol Sanklang, Sanpatong District, Chiang Mai 50120, Thailand; admin@bio-c.co.th; 4Division of Food Science and Technology, Faculty of Agro-Industry, Chiang Mai University, Chiang Mai 50100, Thailand; suphat.phongthai@cmu.ac.th; 5Research Institute for Health Sciences, Chiang Mai University, Chiang Mai 50200, Thailand; sakaewan.o@cmu.ac.th

**Keywords:** angiotensin-I converting enzyme (ACE), silkworm hydrolysate powder, mature Thai silkworm, commercial protease, lowering blood pressure

## Abstract

The angiotensin-I converting enzyme (ACE) plays a pivotal role in hypertension, and while ACE inhibitors are conventional in hypertension management, synthetic medications often carry undesirable side effects. This has spurred interest in alternative ACE inhibitors derived from natural sources, such as edible insects. The silkworm, recognized for its bioactive peptides with potent ACE-inhibitory properties, has emerged as a promising candidate. This study aims to evaluate the acute toxicity and assess the antihypertensive efficacy of crude mature silkworm hydrolysate powder (MSHP) obtained from mature Thai silkworms. Utilizing the commercial protease Alcalase^®^2.4L, MSHP was administered at various doses, including 50, 100, and 200 mg kg^−1^, to hypertensive rats. The investigation spans a 14-day period to observe any potential acute toxic effects. Results indicate that MSHP exhibits LD50 values equal to or exceeding 2000 mg kg^−1^, signifying a low level of acute toxicity. Furthermore, the effective dose for blood pressure reduction in hypertensive rats surpasses 100 mg kg^−1^ of rat body weight. These findings suggest that MSHP derived from Thai mature silkworms holds promise as a natural antihypertensive food source. The implications of this research extend to the development of functional foods, functional ingredients, and dietary supplements aimed at managing hypertension.

## 1. Introduction

Hypertension, frequently known as high blood pressure, manifests when the pressure within an individual’s blood vessels surpasses the critical threshold of 140/90 mmHg. This symptom affected more than 1 billion adults worldwide in 2022 [1]. The angiotensin-I converting enzyme (ACE) belongs to the category of zinc proteases, requiring the presence of zinc and chloride ions for its activation. This enzyme assumes a critical role in the regulation of blood pressure by participating in the renin–angiotensin system [2]. ACE facilitates the conversion of an inactive form of angiotensin-I (decapeptides) into a potent vasopressor, angiotensin-II (octapeptide). The inhibition of ACE activity leads to a diminished formation of angiotensin-II, concurrently accompanied by a decreased degradation of bradykinin, resulting in a reduction in blood pressure [3]. To lower blood pressure through the ACE-related pathway, synthetic ACE inhibitors such as Captopril, Enalapril/Enalaprilat, Benazepril, Fosinopril, Lisinopril, Moexipril, Perindopril, Quinapril, Ramipril, and Trandolapril are commonly prescribed [4]. It is noteworthy that synthesized ACE inhibitor medications are associated with certain undesirable side effects, including cough, hyperkalemia, skin rash, neutropenia, taste disorders, idiosyncratic reactions, and nephritic syndrome [5,6,7,8]. In response to these adverse effects attributed to synthetic medications, there is a rising interest in exploring alternative ACE inhibitors sourced from natural origins. There is particular interest in bioactive peptide hydrolysates derived from edible insects, including desert locust (*Schistocerca gregaria*), mealworm (*Tenebrio molitor*), grasshoppers, red palm weevil (*Rhynchophorus ferrugineus* larva stage), and silkworm (larva and pupa stages) [9,10,11,12,13]

In this study, the focus is on the Thai silkworm (*Bombyx mori* L.). The life cycle of the silkworm initiates with the egg stage, followed by the larval stage, comprising five instars (1st instar to 5th instar), lasting approximately 21 to 25 days. Upon reaching the final phase of the larval stage, the silkworm is referred to as a mature silkworm (MS). The MS represents the conclusive stage of silkworm larval development, occurring just before the spinning of cocoons.

Traditionally, extensive research has been conducted on silkworm pupae, particularly regarding their hydrolysates. Silkworm pupae hydrolysate, acknowledged as a promising source of bioactive compounds, has garnered considerable attention due to its potential health benefits, notably its antihypertensive properties. Notably, studies have extensively explored the ACE-inhibiting potential in laboratory settings [9,10,14,15]. Moreover, recent investigations have expanded beyond in vitro ACE inhibition, revealing that silkworm pupae hydrolysate with a molecular weight of less than 5 kDa can effectively reduce blood pressure in hypertensive rats [16]. This observation underscores a potential research gap concerning hydrolysis. Most studies about ACE inhibition have centered on silkworm pupae and their small peptides or purified peptides, with limited exploration of hydrolysates from mature silkworms and their crude hydrolysates.

The aim of this research is to evaluate crude mature silkworm hydrolysate concerning acute toxicity and its potential for lowering blood pressure in hypertensive individuals. The study’s findings could serve as a reference, suggesting that crude mature silkworm hydrolysate possesses the potency to lower blood pressure in hypertensive rats. Notably, crude mature silkworm hydrolysate derived from mature silkworms incurs a low production cost due to the absence of a molecular separation process. As such, crude mature silkworm hydrolysate should be the candidate for utilization as a bioactive ingredient in the functional food, supplementary products, and pharmaceutical food industries.

## 2. Materials and Methods

### 2.1. Materials

Mature silkworms (MSs) from local mulberry farms in San-Sai District, Chiang Mai Province, Thailand. (18.9031280, 99.0139179) Alcalase^®^2.4L (Novozymes, Bagsvaerd, Denmark); Male BrlHan: WIST@J cl (GALAS) rats (Nomura Siam International Co., Ltd., Bangkok, Thailand); ACE kit-WST (Dojindo Inc., Kumamoto, Japan); and N-Nitro-L-arginine methylester (Sigma-Aldrich, Burlington, MA, USA).

### 2.2. Sample Preparation

For the preparation of mature silkworm powder, 4 kg of boiled mature silkworms (MSs) were subjected to drying in a vacuum microwave oven (VMO) using a 6400-magnetron watt/kg sample (SAMSUNG, Suwon, Republic of Korea) at vacuum pressure (−750 mmHg). The drying process concluded when the moisture content in the MSs dropped below 10% (approximately 3 h). Subsequently, the dried MS samples were pulverized into a fine powder, referred to as mature silkworm powder (MSP), using a blender (Nanotech, China) operating at 25,000 rpm, for further analysis.

For the preparation of crude mature silkworm hydrolysate powder, briefly, 5 g of MSP was subjected to hydrolysis using 5% of the commercial enzyme Alcalase^®^2.4L (EC 3.4.21.14) (Activity 2.54 AU-A/g) (Novozymes, Denmark) (Ingredient of enzyme consisted of Glycerol, CAS no. 56-81-5, water, CAS no. 7732-18-5 and protease (Subtilisin), CAS no. 9014-01-1*) in 0.1 mM phosphate buffer pH 8 (ratio MSP to phosphate buffer is 1:10 %*w*/*v*). The mixture was incubated with the temperature controlled at 60 °C for 1 h. Subsequently, the hydrolysate mixtures were heated up to 95 °C for 10 min in a water bath to deactivate the enzymes. After that, the mixture was cooled to ambient temperature, then centrifuged at 1008× *g* for 30 min. The supernatants of the mixture were collected and freeze-dried (Christ: Alpha 1-4 LSC plus, Burladingen, Germany) to obtain crude mature silkworm hydrolysate powder (MSHP), and kept at −20 °C for acute toxicity study, determination of IC_50_, and anti-hypertension in the hypertensive rat.

### 2.3. Animal Study for Acute Toxicity Assessment

Male BrlHan: WIST@J cl (GALAS) rats (12 weeks, weight, 309 ± 22.1 g) were used. The protocol was approved by the committees at the Faculty of Medicine, Chiang Mai University, Chiang Mai, Thailand (Permit Number: 34/2565). All rats (10 rats: 5 rats in the control group and 5 rats in the experimental group) were housed at a controlled temperature (25 ± 1 °C) and lighting in a 12:12-h-light/dark cycle with food and water provided ad libitum. The acute toxicity test was performed according to OECD Test Guidelines 420 (Fixed Dose Procedure) for five days. Both the experimental and control group rats underwent a 12 h-fasting period before commencing the testing procedures. For the experimental group, a dose of 2000 mg kg^−1^ body weight of silkworm hydrolysate powder (MSHP) was administered to five male rats based on their body weights. For the control group, deionized water was fed at a similar dose to the experimental group. The rats were closely monitored for the initial 30 min and observed for an additional 4 h after the feeding. The normal chow diet was then provided after the survival of the treated rat. All the groups were further observed for any potential toxic effects for 14 consecutive days. At the end of the study, the rats were subjected to euthanasia with an overdose of thiopental sodium (250 mg kg^−1^). Vital organs (liver, kidney, heart, and spleen) were excised, weighted, and preserved in 10% formalin for histopathology study.

### 2.4. Determination of IC_50_ ACE-Inhibition of Silkworm Hydrolysate Powder

The angiotensin-converting enzyme (ACE) inhibition of silkworm extracts was determined according to Iwamoto et al., 2023 [17] following the colorimetric method using the ACE kit-WST (Dojindo Inc., Kumamoto, Japan) Briefly, 2 mg mL^−1^ of silkworm hydrolysate and Enalapril (ACE-inhibitor used as the positive control) underwent a series of 5-fold dilutions to produce concentrations of 2 mg mL^−1^, 0.4 mg mL^−1^, 0.08 mg mL^−1^, 0.016 mg mL^−1^, and 0.0032 mg mL^−1^. Subsequently, 20 µL of silkworm hydrolysates, Enalapril, or deionized water (used as a control and blank) was added to each well in a 96-well plate. After that, 20 µL of substrate buffer was added to each well, and then 20 µL of enzyme working solution was added to each sample and control well (for the blank well, deionized water was added). The plate underwent incubation at 37 °C for 1 h. After this, 200 µL of the indicator working solution was introduced into each well containing the reagent solutions, with subsequent incubation at 37 °C for 10 min. The absorbances of the samples, positive control, control, and blank were then recorded at 450 nm using a microplate reader (SPECTROstar Nano, BMGLABTECH, city, Germany). The ACE inhibition was calculated using the equation:$$\mathrm{ACE}\;\mathrm{inhibition}\;(\%)=(\frac{Absorbance\;of\;control-Absorbance\;of\;sample}{Absorbance\;of\;control-Absorbance\;of\;blank})\times 100$$
where *Absorbance of control* is the absorbance of a solution consisting of deionized water, substrate buffer, enzyme working solution, and indicator working solution (without samples or ACE inhibitor), *Absorbance of the blank* is the absorbance of a solution consisting of deionized water, substrate buffer, and indicator working solution reagent blank (without enzyme working solution).

### 2.5. Anti-Hypertension Treatment in the Hypertensive Rat

The methodology employed for hypertension management in the experimental hypertensive rat study was modified from Pechanova et al. (2020). [18] Male BrlHan: WIST@Jcl (GALAS) rats (10 weeks, 252.8 ± 7.3 g) were used (six rats/group). All rats were housed under a controlled temperature (25 ± 1 °C) and lighting in a 12:12-h-light/dark cycle with food and water ad libitum. They were fed a standard chow diet (product food No. CP082, Perfect Companion Group Co., Ltd., Bangkok, Thailand). The protocol was approved by the committees at the Faculty of Medicine, Chiang Mai University, Chiang Mai, Thailand (Permit Number: 34/2565). After 1 week of acclimatization, the animals were randomly divided into two groups: normal control (NC) and N-Nitro-L-arginine methyl ester (L-NAME) treated group. The rats in the normal control group (NC) received distilled water (vehicle), whereas rats in the hypertensive group were administered L-NAME (40 mg kg^−1^/day) orally until the end of experiment (7 week). After three weeks of L-NAME administration, the L-NAME treated rats were randomly assigned into five subgroups and daily treated of silkworm hydrolysate powder for a further 4 weeks with various types of test samples, consisting of:(a)Hypertensive group (HT): the negative control group that received deionized water.(b)Positive control group (HT-D): received Captopril (25 mg kg^−1^/day).(c)Low dose group (HT-L): received MSHP at a dose of 50 mg kg^−1^/day.(d)Medium dose group (HT-M): received MSHP at a dose of 100 mg kg^−1^/day.(e)High dose group (HT-H): received MSHP at a dose of 200 mg kg^−1^/day.

The measurement of systolic blood pressure (SBP), diastolic blood pressure (DBP), and heart rate (HR) was performed by using an MRBP Mouse and Rat Tail Cuff Method Blood Pressure Systems (IITC Life Science, Woodland Hills, CA, USA) after warming the rats in a warm holder kept at 37–39 °C for 10 min. To minimize stress-induced variations in blood pressure, all measurements were taken by the same person in the environment at the same time of the day. Rats were euthanized with an overdose of thiopental sodium (250 mg kg^−1^) for blood and organ sampling at the end of 7 weeks. Vital organs (liver, kidney, heart, and spleen) were excised and weighed.

### 2.6. Statistical Analysis

The results are expressed as the mean ± standard deviation (SD) derived from three independent experiments. Statistical analyses of the results were conducted using one-way ANOVA with SPSS version 20, including the results of the acute toxicity assessment. In regard to the rats’ organ weight and anti-hypertension treatments in hypertensive rats, all data were calculated using a completely randomized design (CRD) with *p* < 0.05 considered the threshold for significance. The data were further analyzed using Duncan’s triplicates range test to identify significant differences.

## 3. Results

### 3.1. Acute Toxicity Assessment

The results showed that the weight of rats in both the experimental and control groups increased over time, but the differences between the two groups were not statistically significant (*p* > 0.05) (Figure 1). The regular observation of the rats did not reveal any notable alterations in general behavior. Furthermore, there was no recorded mortality among the treated rats. Internal organ examinations and their histopathology results did not reveal any abnormal conditions, and the weights of these organs remained consistent when compared to the control group (Table 1).

### 3.2. Anti-Hypertension Treatment in the Hypertensive Rat

The recorded weights of the rats following L-NAME treatment revealed a consistent upward trend over seven weeks for all groups, including the negative control (NC), hypertensive control (HT), positive control (HT-D), low-dose MSHP treatment (HT-L), medium-dose MSHP treatment (HT-M), and high-dose MSHP treatment (HT-H) (Figure 2). During the study, hypertension was induced by administering L-NAME at a dosage of 40 mg kg^−1^/day throughout the experiment (7 weeks). The results revealed a significant elevation (*p* < 0.001) in both systolic and diastolic blood pressure values in comparison to the control group of rats with normal blood pressure (NC). Specifically, by the end of the study (week 7), rats receiving L-NAME without any treatment exhibited an average systolic blood pressure of up to 206 ± 4.6 mmHg, whereas the control group maintained an average systolic blood pressure of 121 ± 2.5 mmHg.

Crude mature silkworm hydrolysate powder, which was used in this experiment, had an IC_50_ value of 17.75 ± 0.92 µg (IC_50_ of Enalapril as an ACE-blocker was about 0.1 µg). The influence of MSHP on hypertensive rats became notably apparent by the seventh week. Notably, the administration of medium-dose MSHP (HT-M), high-dose MSHP (HT-H), and positive control treatment (HT-D) resulted in systolic/diastolic blood pressure readings of 180 ± 5.2/136 ± 5.3 mmHg, 167 ± 2.6/128 ± 5.2 mmHg, and 147 ± 6.3/117 ± 3.7 mmHg, respectively (Figure 3a and Figure 4a). This demonstrated a significant reduction in both systolic and diastolic blood pressures compared to the hypertensive control group (HT), which recorded readings of 206 ± 4.6/155 ± 5.1 mmHg (*p* < 0.05). When compared to the baseline measurements at week 0, the HT-M, HT-H, and HT-D rat groups exhibited significant reductions in systolic/diastolic blood pressure at week 7, with values of 60/49, 50/40, and 28/22, respectively (*p* < 0.05) (Figure 3b and Figure 4b.)

The result of this experiment suggests that crude mature silkworm hydrolysate powder from mature silkworm powder had bioactive peptides with the potent potential to inhibit Angiotensin-Converting enzyme activity, thereby possibly contributing to a reduction in blood pressure among hypertensive rats.

Regarding the results of the organ weights, the findings indicate that rats in the hypertensive group (HT) and rats in the positive control group (HT-D) exhibited higher heart weights compared to rats in the normal control group (NC) (*p* < 0.05). Conversely, rats receiving silkworm hydrolysate powder (MSHP) at low (HT-L), middle (HT-M), and high doses (HT-H) demonstrated heart weights that were not significantly different compared to the NC group (*p* > 0.05). In contrast, the analysis of organ weights, including the liver, kidneys, and spleen, revealed no significant differences (*p* > 0.05) among all six groups (NC, HT, HT-L, HT-M, HT-H, and HT-D) (Table 2).

## 4. Discussion

### 4.1. Acute Toxicity Assessment

The acute toxicity assessment serves as the primary method for verifying toxicity to determine the maximum safe dosage for short-term consumption. The safety dose determined from acute toxicity analysis can be utilized to establish a maximum baseline for further investigations, including assessments for anti-hypertensive effects in hypertensive rats, sub-chronic toxicity, and chronic toxicity. After a 14-day testing period, the results of the mature silkworm hydrolysate powder (MSHP) study indicated that internal organ examinations did not reveal abnormal conditions, and the weights of these organs remained consistent. The organ weight results align with those obtained from the hydrolyzed collagen sourced from *Lates calcarifer* skin [19]. Furthermore, no instances of mortality were recorded throughout the entire 14-day experimental duration. This result exhibited a similar trend to that observed in rats administered protein hydrolysate obtained from microalgal and casein hydrolysate, which had anti-hypertensive properties [20,21]. This acute toxicity of MSHP was similar trend to silkworm pupa powder [22] The administration of MSHP at a dosage of 2000 mg/kg of rat body weight was interpreted as involving substances with low or non-toxicity in terms of acute toxicity.

Therefore, in the short-term period, SPH hydrolyzed via Alcalase^®^2.4L was safe for consumption, similar to whey protein concentrate, isolated peptides from whey protein, and palm kernel cake hydrolysate [23,24,25]. The advantage of this experiment lies in its focus on the short-term adverse effects of silkworm hydrolysate powder when administered in a single dose. The results of the acute toxicity study serve as a reference for determining the appropriate sample amount in subsequent animal studies. The limitation of acute toxicity lies in its short-term nature. However, to ascertain the toxicity profile of silkworm hydrolysate, it is imperative to assess sub-chronic and/or chronic toxicity, ensuring a comprehensive understanding of its long-term effects.

### 4.2. Anti-Hypertension in the Hypertensive Rat

In this study, hypertensive conditions in rats were induced using Nitro-L-arginine methyl-ester (L-NAME). L-NAME is a chemical known for inhibiting nitric oxide (NO) production, leading to arterial hypertension in rats [26]. When nitric oxide (NO) levels decrease, there is a consequential elevation in both systolic and diastolic blood pressure values. This phenomenon is attributed to the activity of angiotensin-I converting enzyme (ACE) (EC 3.4.15.1). ACE is a category of zinc proteases, necessitating the presence of zinc and chloride ions for its activation. This enzyme plays a pivotal role in regulating blood pressure through its involvement in the renin–angiotensin system. ACE serves the crucial function of converting the inactive decapeptide by releasing a C-terminal dipeptide, characterized as an oligopeptide †. Xaa-Yaa occurs when Xaa is not Pro and Yaa is neither Asp nor Glu. This mechanism could change angiotensin I into the potent vasoconstrictor octapeptide known as angiotensin II [26]. Additionally, ACE can deactivate the antihypertensive vasodilator bradykinin. Consequently, the inhibition of ACE emerges as a significant therapeutic approach for the management of hypertension [10,27].

The results of the anti-hypertensive effects in hypertensive rats suggest that those rats receiving crude hydrolysate (MSHP) at doses of 100 and 200 mg/kg exhibited a reduction in blood pressure, with systolic/diastolic readings of 180 ± 5.2/136 ± 5.3 mmHg and 167 ± 2.6/128 ± 5.2 mmHg, respectively, compared to the hypertensive control group (206 ± 4.6/155 ± 5.1 mmHg). Previous research on silkworm pupa has demonstrated a reduction in blood pressure in hypertensive rats through the administration of isolated hydrolysates, particularly those with doses of 80 mg kg^−1^ per rat body weight and a molecular weight (MW) of less than 5 kDa [28].

One plausible mechanism behind this observed effect is that crude mature silkworm hydrolysate may contain glutamic acid at the C-terminal of the peptide chain and hydrophobic residues at the N-terminal, which could contribute to a significant reduction in systolic blood pressure [3]. Furthermore, it is postulated that peptide chains present in MSHP may retain ACE-inhibition properties due to resistance against processes such as intestinal degradation, plasma peptidase breakdown, or digestion within the rat’s digestive tract [12]. Another proposed mechanism is the binding of peptides to the substrate-binding site of ACE through hydrogen bonds, akin to the action of ACE-blocker medications [9]. Additionally, various physiological mechanisms, including the autonomic nervous system, endothelin system, and nitric oxide system, are known contributors to high blood pressure [16]. Considering these multifaceted factors, it can be inferred that crude mature silkworm hydrolysate may effectively reduce blood pressure in hypertensive rats by modulating these physiological mechanisms.

From the results, a dose-dependent relationship was observed, indicating that an increase in MSHP dosage corresponded to a more pronounced effect on reducing both systolic blood pressure (SBP) and diastolic blood pressure (DBP). This phenomenon aligns with findings in hypertensive rats that received pea protein hydrolysate and whey protein hydrolysate [16,29]. The noteworthy aspect in this context lies in the potential exhibited by the crude, mature silkworm hydrolysate powder. Traditionally, previous research has identified novel bioactive peptides in silkworm pupae that exhibit potent ACE inhibition, including Trp-Trp, Gly-Asn-Pro-Trp-Trp, Asn-Pro-Trp-Trp, Pro-Trp-Trp [9], APPPK [10], Ala-Ser-Leu [14], Lys-His-Val [30], and Val-Glu-Ile-Ser [31].

In terms of rat organ weight, the results revealed that the heart weight of rats in the hypertensive group and positive control group was higher than that of the other groups. This observation may be attributed to the influence of L-NAME, as it was found to increase both cardiac and aortic weights. Furthermore, the histological analysis indicated an increase in the thickness of the vascular wall and fibroblast infiltration in the myocardium. The elevation in cardiac weight, facilitated by an increase in fibroblasts, suggests that L-NAME could induce cardiac hypertrophy by activating the renin–angiotensin system [26,32]. Rats treated with MSHP exhibited a lower heart weight compared to the HT and HT-D groups. This result could be attributed to the potential of MSHP to alleviate the effects of L-NAME. This observation aligns well with previous findings on *Cinnamomum zeylanicum* stem bark extract [25].

The limitation of the anti-hypertensive effect observed in hypertensive rats is attributed to animal studies. Additionally, questions arise regarding the applicability of the effects of silkworm hydrolysate powder on human subjects. Consequently, future research endeavors should encompass clinical trials to address these questions and assess the impact of a human study.

Based on the results of this experiment, it can be concluded that the silkworm hydrolysate powder obtained from mature Thai silkworms is a promising candidate as an alternative natural ACE inhibitor. The crude, mature silkworm hydrolysate powder demonstrates effectiveness in lowering blood pressure. Future research will focus on investigating sub-chronic toxicity to ensure the safety of crude, mature silkworm hydrolysate powder for daily supplementary use. Furthermore, if the silkworm hydrolysate powder proves to be safe in terms of sub-chronic toxicity or chronic toxicity, its potential for lowering blood pressure could be further evaluated in a clinical trial.

## 5. Conclusions

In conclusion, the acute toxicity assessment and anti-hypertension study in hypertensive rats support the efficacy of crude mature Thai silkworm hydrolysate powder produced via Alcalase^®^2.4L. The LD_50_ values equal to or exceeding 2000 mg kg^−1^ Furthermore, the effective dosage for lowering blood pressure in hypertensive rats was found to be equal to or exceeding 100 mg kg^−1^ of the rat’s body weight. These findings align with the research objectives, establishing crude, mature Thai silkworm hydrolysate powder as a viable candidate for functional food, functional ingredient, or supplement applications in addressing antihypertensive concerns. Future research will focus on sub-chronic/chronic toxicity studies to confirm the long-term safety of crude mature Thai Thai silkworm hydrolysate powder. If the results support the safety of prolonged use, further investigations can assess the hydrolysate’s efficacy in reducing blood pressure through clinical trials.

## Figures and Tables

**Figure 1 foods-13-00943-f001:**
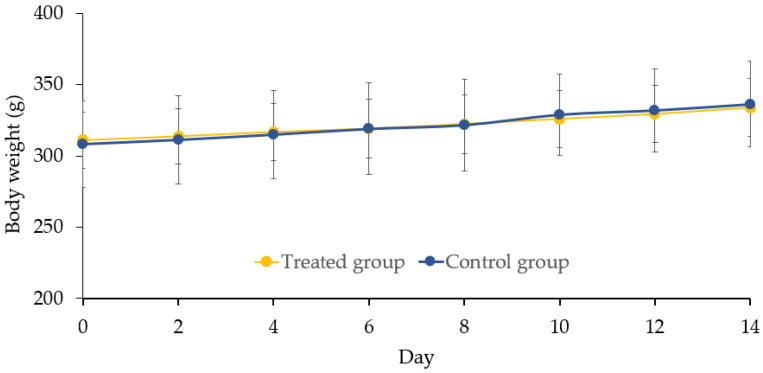
Rat body weight changes in the two groups over 14 days.

**Figure 2 foods-13-00943-f002:**
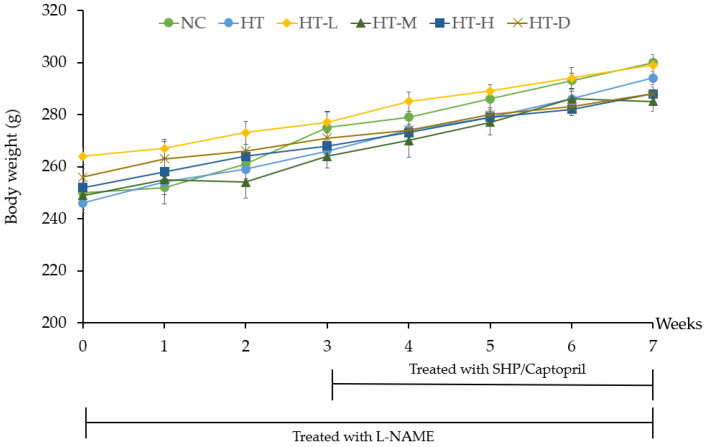
Mean weekly body weight of the rats over the course of 7 weeks. Note: NC is rats in the normal control group, HT is hypertensive rats that received deionized water, HT-L is the rats that received 50 mg kg^−1^ body weight (low-dose) of silkworm hydrolysate powder, HT-M is the rats that received 100 mg kg^−1^ body weight (middle-dose) of silkworm hydrolysate powder, HT-H is the rats that received 200 mg kg^−1^ body weight (high-dose) of silkworm hydrolysate powder, and HT-D is the rats that received Captopril 25 mg kg^−1^ body weight.

**Figure 3 foods-13-00943-f003:**
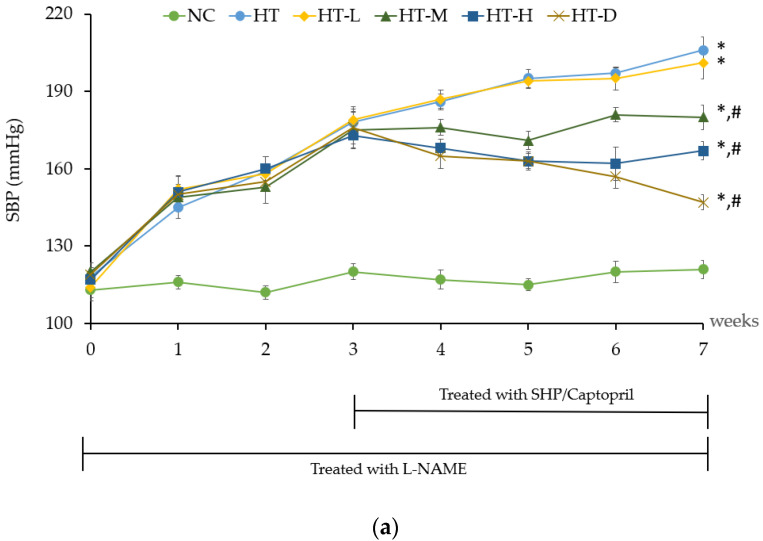
Effect of hydrolysate powder on the systolic blood pressure of hypertensive rats (**a**) and %change in systolic blood pressure compared with the normal control group (**b**). Note: NC is rats in the normal control group, HT is hypertensive rats that received deionized water, HT-L is the rats that received 50 mg kg^−1^ body weight (low-dose) of silkworm hydrolysate powder, HT-M is the rats that received 100 mg kg^−1^ body weight (middle-dose) of silkworm hydrolysate powder, HT-H is the rats that received 200 mg kg^−1^ body weight (high-dose) of silkworm hydrolysate powder, and HT-D is the rats that received Captopril 25 mg kg^−1^ body weight. * is significantly different when compared to NC group and # is significantly different when compared to the HT group.

**Figure 4 foods-13-00943-f004:**
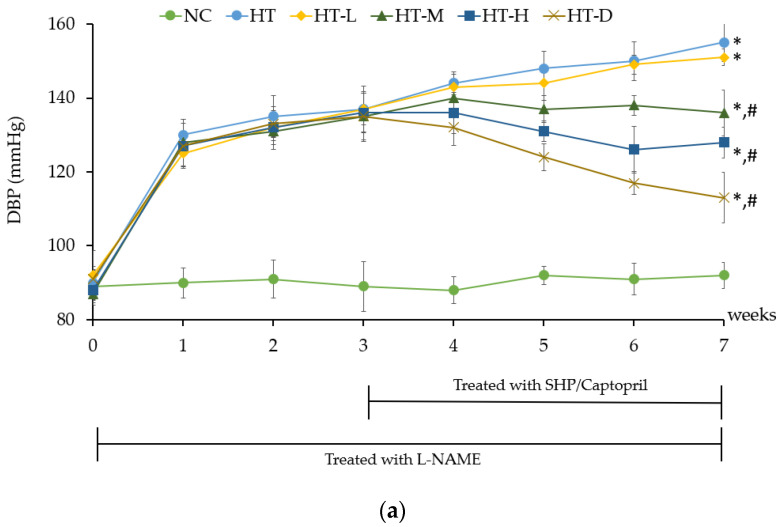
Effect of hydrolysate powder on diastolic blood pressure (DBP) of hypertensive rats (**a**), %change in DBP compared with the normal control group (∆DBP) (**b**). Note: NC is rats in the normal control group, HT is hypertensive rats that received deionized water, HT-L is the rats that received 50 mg kg^−1^ body weight (low-dose) of silkworm hydrolysate powder, HT-M is the rats that received 100 mg kg^−1^ body weight (middle-dose) of silkworm hydrolysate powder, HT-H is the rats that received 200 mg kg^−1^ body weight (high-dose) of silkworm hydrolysate powder, and HT-D is the rats that received Captopril 25 mg kg^−1^ body weight. * is significantly different when compared to NC group and # is significantly different when compared to the HT group.

**Table 1 foods-13-00943-t001:** Organ weight of the rats after 14 days of study.

Organs	Control Group ^ns^	Experimental Group ^ns^
Liver	10.24 ± 1.22	11.42 ± 0.72
Kidney (both sides)	2.12 ± 0.12	2.45 ± 0.11
Heart	0.91 ± 0.06	0.92 ± 0.02
Spleen	0.54 ± 0.05	0.66 ± 0.07

Data are presented as the mean ± standard deviation (SD). ns: non-significantly different between the two groups.

**Table 2 foods-13-00943-t002:** Organ weight of rats fed silkworm power for 3 weeks.

Rat Group	Weight of Organs				
	Spleen	Heart	Liver	Kidney (Left)	Kidney (Right)
NC	0.72 ± 0.04	0.80 ± 0.05	11.14 ± 1.26	1.20 ± 0.32	1.21 ± 0.20
HT	0.65 ± 0.03	0.95 * ± 0.06	10.89 ± 0.86	1.14 ± 0.02	1.16 ± 0.09
HT-D	0.65 ± 0.04	0.90 * ± 0.03	11.32 ± 0.88	1.16 ± 0.08	1.13 ± 0.10
HT-L	0.61 ± 0.14	0.89 ± 0.12	10.23 ± 2.03	0.98 ± 0.08	1.05 ± 0.11
HT-M	0.63 ± 0.11	0.89 ± 0.11	11.19 ± 1.81	1.13 ± 0.04	1.13 ± 0.05
HT-H	0.64 ± 0.05	0.85 ± 0.07	11.82 ± 1.35	1.06 ± 0.11	1.08 ± 0.13

Abbreviations: NC = normal control, HT = hypertensive group, HT-D = positive control, HT-L = silkworm hydrolysate powder low dose, HT-M = silkworm hydrolysate powder medium dose, and HT-H = silkworm hydrolysate powder high dose, Data are presented as the mean ± standard deviation (SD) from 5 rats, * *p* < 0.05 when compared to the normal control group.

## Data Availability

The original contributions presented in the study are included in the article, further inquiries can be directed to the corresponding author.

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
