# Peer review of "Evaluation of Thai Silkworm (Bombyx mori L.) Hydrolysate Powder for Blood Pressure Reduction in Hypertensive Rats"

_foods, 2024, doi:10.3390/foods13060943_

Round 1

Reviewer 1 Report

Comments and Suggestions for Authors

The study submitted for our analysis cannot be published in this form. It lacks fundamental knowledge on the used enzymes and their mechanisms of action. My observations are the followings:

- the title should be shorted because it is too long.

he introduction does not allow an easy understanding of the research hypothesis because they are not  clear. It should therefore be revised. We think that the others should focus the potential sources of hypertension, hence the interest in dealing with these causes. To this end, the side-effects associated with these drugs are encouraging research into new plant-based nutraceuticals. Hence the need to verify their toxicity and efficacy by various mechanisms (ACE inhibition, blood pressure reduction, diuretic activity, etc). The biochemical content of these plants could be influenced by biotic stresses, the season, the zone, the parts of the plant, etc., it is important to be aware of these factors. It would be interesting to combine this with biochemical characterization, which would probably justify the biological effectiveness of the extracts.

-The central role and mechanism of action of angiotensin-converting enzyme (EC 3.4.15.) is weakly explained.

It is a peptidyl-dipeptidase A. Reaction: Release of a C-terminal dipeptide, oligopeptideXaa-Yaa, when Xaa is not Pro, and Yaa is neither Asp nor Glu. Thus, conversion of angiotensin I to angiotensin II, with increase in vasoconstrictor activity, but no action on angiotensin II!!!

See detail of the enzyme:  https://www.enzyme-database.org/query.php?ec=3.4.15.1

-          Alcalase® 2.4 L FG is a serine endo-peptidase, please like all the used or cited enzymes, give their EC numbers. If it a bulk commercial proteases, you should give names of proteases that are in the commercial preparation.

-          -There are no transitions between the stated ideas, hence the lack of fluidity in the writing style.

The methodology lacks numerous bibliographical references.

Other remarks:

Line 38: Please delete the word "above" as you have already mentioned superior or simply reformulate the sentence

Line 38: Please specify the year of these statistics

Verifying the safety of a plant extract would require at least the assessment of sub-acute activity.anti-hypertensive drugs are taken on an ongoing basis to control hypertension, not as a single dose. This poses a major problem in terms of research methodology. Better still, the anti-hypertensive activity was assessed by daily gavage of the extracts after 4 weeks, not as a single dose. Consequently, if safety is an objective of the study, the authors must evaluate subacute activity.

No reference was given for the preparation of the extract. This explains the confusion in the extraction procedure. The authors claim to have freeze-dried extracts which were heated to 95°C. What is the point of this lyophilisation? With regard to this example, there are many contractions in this paragraph. It should be taken up again to give us a better idea of where we stand.

Why use rats of only one sex? It would be better to use both sexes and the minimum number of rats should be 6.

Or use female rats or mice.

The authors claim to have carried out histopathological analyses on line 105. These results do not appear in the manuscript.

Please provide the reference for the methodology used to assess the IC50 for angiotensin-converting enzyme inhibition.

Figure 1 should be repeated, using different colours for the control group and the test group. This makes the graph easier to see and understand.

In Table 1, we suggest that the superscript "ns" be used in the columns for the control group and the test group.

Figure 2 (a) and 3 (a) need to be improved, using different colours for the different groups. This will make the graph easier to see and understand.

The discussion is poor and needs to be improved. The authors should compare their results with previous studies on plants or similar themes on other species. They should also explain why this plant deserves particular interest. We suggest that the authors carry out biochemical characterizations to make this discussion more interesting while explaining the probable mechanism of action of these extracts.

The conclusion needs to be improved. This is the place for the authors to tell us whether the objectives of the study have been achieved, not a summary of the results as in the summary. The conclusion it is not well supported by the initial idea.

Comments on the Quality of English Language

The English is roughly acceptable.

Author Response

Dear reviewer

Thank you sincerely for your invaluable suggestions. I have carefully incorporated your feedback into my manuscript, and I am pleased to share the revised version with you. Please see the attachment.

Reviewer 2 Report

Comments and Suggestions for Authors

There are major issues with the manuscript. I have attached all my comments in the PDF to improve the manuscript. Also, the English should be improved.

Comments on the Quality of English Language

Major revision

Author Response

(The authors gave the same response as above.)

Reviewer 3 Report

Comments and Suggestions for Authors

1- line 111 please delete the repeated mg/mL

2-Please add the active ingredient found in  silkworm

3-please add the  results in IC50

4- figure 1body weight change in two groups 14 days in methods 3 weeks, please correct

Author Response

(The authors gave the same response as above.)

Reviewer 4 Report

Comments and Suggestions for Authors

Dear authors,

The manuscript entitled "Assessment of the Safety and Efficiency of Thai Silkworm (Bombyx mori L.) Hydrolysate Powder to Lowering Blood Pressure in Hypertensive Rats" evaluate the the acute toxicity of silkworm hydrolysate powder (SHP) obtained from Thai mature silkworms using a commercial protease (Alcalase®2.4L) and its effectiveness in lowering blood pressure in hypertensive rats. It presents scientific relevance for Health, Medicine, Pharmacy and others area. The language (English) is satisfactory (but, I suggest the final revision)! However, you need to change some details/information in the Abstract, Introduction, Material and Methods, results and discussion, and conclusions.

1. Abstract: Adequate! But:

- The abstract is well written, but without many details of the methods used! I suggest inserting the methods parameters and results obtained (numerical data!!!) more relevant.

- In lines 27-28: To replace “mg/kg” by “mg Kg-1”, and throughout the manuscript (including all tables/figures).

- At the end, I suggest highlighting the advantages/ disadvantages of the study and methods.

2. Introduction section:

- I suggest expanding the text a little further, including information about Thai Silkworm (Bombyx mori L.), methods and analysis techniques used in the study.

3. Materials and methods section: The methodological proposal is appropriate to the manuscript, but I suggest:

- Page 2, in “2.1. Materials” section: What are the conditions for collection/acquisition and storage of samples? What is the time/period (from acquisition to analysis)?

- Page 2, in “2.2. Sample preparation” section: I suggest indicating the references of the protocols used for preparation of mature silkworm (hydrolyzed and powder).

- Pages 2-3, in “2.3. Animal Study for Acute Toxicity Assessment” section: The authors wrote: “Male BrlHan: WIST@J cl (GALAS) rats (12 weeks, weight, 309 ± 22.1 g) were used…”. What is the preference for this species? Could another one be used? And why only males? I suggest highlighting this information in the manuscript. What protocol/reference is used for such procedures? I suggest including references!

- Page 3, in “2.4. Determination IC50 ACE-Inhibition of silkworm hydrolysate powder” section: What protocol/reference is used for such procedures? I suggest including references! To replace “mg/mL” by “mg mL-1”, and throughout the manuscript (including all tables/figures).

- Pages 3-4, in “2.5. Anti-hypertension in the hypertensive rat” section: What protocol/reference is used for such procedures? I suggest including references! In lines

4. Results section (or “Results and discussion”)?

Wouldn't it be more interesting to combine the "results” with the "discussion" to better describe the findings and compare them with other works published in the literature?? I suggest expanding the discussions!

- Pages 8-9, in “4.1. Acute toxicity assessment” and “4.2. Anti-hypertension in the hypertensive rat” sections: I suggest expanding the discussions - better describe the findings and compare them with other works published in the literature!

- I suggest, at the end of the "results and discussion", to write a paragraph summarizing the findings and their impacts on the research proposal.

5. Conclusion: suggest pointing out the main results and disadvantages/limitations of the method and the study!

6. Table and Figures: Adequate. To review the units of measurements/concentrations!

7. References: Please, check if the references are in accordance with the journal's rules.

Comments on the Quality of English Language

The language (English) is satisfactory (but, I suggest the final revision)!

Author Response

(The authors gave the same response as above.)

Round 2

Reviewer 1 Report

Comments and Suggestions for Authors

The article may be acceptable after minor editing errors

Comments on the Quality of English Language

The English acceptable but can be improved.

Reviewer 2 Report

Comments and Suggestions for Authors

accept

Comments on the Quality of English Language

minor

Reviewer 3 Report

Comments and Suggestions for Authors

Accept in present form